# Physical Demands during the Game and Compensatory Training Session (MD + 1) in Elite Football Players Using Global Positioning System Device

**DOI:** 10.3390/s22103872

**Published:** 2022-05-19

**Authors:** Gabriel Calderón-Pellegrino, Leonor Gallardo, Jorge Garcia-Unanue, Jose Luis Felipe, Antonio Hernandez-Martin, Víctor Paredes-Hernández, Javier Sánchez-Sánchez

**Affiliations:** 1IGOID Research Group, University of Castilla-La Mancha, 45071 Toledo, Spain; leonor.gallardo@uclm.es (L.G.); jorge.garciaunanue@uclm.es (J.G.-U.); antonio.hmartinsan@uclm.es (A.H.-M.); 2Faculty of Sport Sciences, Universidad Europea de Madrid, 28670 Madrid, Spain; joseluis.felipe@universidadeuropea.es (J.L.F.); javier.sanchez2@universidadeuropea.es (J.S.-S.); 3Sport Science Institute, Universidad Camilo José Cela, 28692 Madrid, Spain; vparedes@ucjc.edu

**Keywords:** global positioning system, tracking system, quantification, substitutes, high-intensity, compensatory training

## Abstract

The aims of this study were to analyze the differences of physical demands of non-starter players regarding the playing time during the competition and to evaluate the physical demands of the compensatory training (MD + 1C) for substitute players in elite football. The match statistics and MD + 1C of substitute players from a professional Spanish LaLiga football club were analyzed using a 10-Hz global positioning system (GPS) Apex GPS system device, which has been validated as a reliable and valid method to analyze performance in team sports, during all games of the 2016/2017, 2017/2018 and 2018/2019 seasons. The starting players showed both lower total distances covered and high-intensity actions compared to the substitutes. Regarding the minutes played by the substitutes, greater physical performance was found for the players with fewer minutes (5–15 min). Furthermore, no differences were found between first and second divisions regarding physical performance of substitutes (*p* > 0.05). This study highlights the importance of individualizing the workload of training sessions for substitutes and starters. Furthermore, the complementary session should be individualized according to the minutes played by the substitutes. These players are potentially under-loaded compared to starters, especially in terms of high-intensity actions, therefore additional session-specific training for each substitute would be useful to reach the optimal training load according to the minutes played during the game.

## 1. Introduction

Football is an intermittent sport which alternates different physical actions such as walking, jogging and running at low, medium and high intensities [1]. High-intensity actions (HIAs) and sprints are considered as one of the most important activities in the match and are associated with the most frequent actions in goal situations [2,3,4].

Global positioning system (GPS) devices are commonly used in several team-sports and help coaches to assess specific movement demands of the players during training sessions and competitions [5]. Many studies of the reliability and validity of GPS in team sports have been made from 2009 [6]. However, there is limited information regarding the validity and reliability of these devices during high-intensity actions [7]. The main findings were that it is a valid method to measure distances at low speed, however, its validity seems to be affected by path linearity and movement intensity [8]. This technology has been demonstrated as a valid and reliable method to analyze the performance across a match, between matches and level of competition, mostly at low intensity actions [6]. More specifically, Coutts and Duffield [7] noticed that GPS has an acceptable level of accuracy and reliability for performance parameters relevant to team sports such as total distance and peak speeds during high-intensity actions, brief and intermittent exercises over a non-linear course.

Performance in HIAs can be greatly different between starters and substitute players [9]. Substitutions during a match are allowed by football rules and usually coaches execute it to change tactics, replacing players who are underperforming or injured, giving playing opportunity to young players or players returning from injury [10].

During the game, substitutes should be able to perform at higher intensities than starters [11]. It has been demonstrated that substitutes covered 25% more HIAs and 63% greater sprinting distances during the final 15 min of a game, compared to starters over the same period [12]. Mean second-half heart rate was significantly higher (84 ± 3 vs. 81 ± 4% maximum heart rate) in substitutes compared with players who completed 90 min [13]. Moreover, substitutes covered greater total distance (TD) and 10% more high-intensity running (HIR), with full-backs being the only position for whom substitutes’ HIR did not exceed that of players being replaced [14].

The competition game has been quantified as the most demanding session of the week [15]. An important number of players in the team are not exposed to the total training load (TL) of the game [9]. In this line, Kraemer et al. [16] found that physical fitness performance decreased in non-starter players during the season due to the lack of exposure to competition. For that reason, to try to approach the game’s training load experienced by the starters, substitutes need a complementary training session (MD + 1C) [17].

However, it has been demonstrated that MD + 1C training load carried out by non-starters was substantially lower than the magnitude produced by the competition game [9]. Moreover, all training load variables (total distance, energy expenditure, time spent above 90% HR_max_, accelerations, decelerations and high-intensity running) of this session were significantly lower than regular training on match day −4 (4 days before the competition), which is considered the most demanding session of the week [17]. These sessions are composed of a smaller number of players (~9 vs. ~18 in regular training) and an increase in ball touches, dribbles and duels, but lower physical demands [18]. Thus, even though substitutes performed a complementary training session to compensate the absence of participation during the competition, weekly training load for the starters was largely higher than in the non-starters [9].

During a week with one match, non-starters on average showed a lower total load than starters (up to ~30% less for running and high-speed running) [17]. Moreover, starters accumulated greater (large/very large) perceived training load than the non-starters, the official matches being the source of such differences, as this highlights the general risk of underloading non-starters [9].

For these reasons, for substitutes players it is necessary to have compensatory training sessions with either higher intensity, volume or both, organize additional friendly games or let them play in matches for a lower team [17]. These volume and intensity data need to be collected by GPS with high frequency rate because it provides greater validity to measuring physical parameters such as distance and speed [19]. A 10-Hz GPS demonstrated a lower standard error over a 15 m sprint comparing with 5-Hz and 1-Hz GPS [6].

Previous studies have analyzed the physical demands of non-starters, however none of them have separated the minutes of play of each substitute. Therefore, the purpose of this study was to analyze the differences of physical demands of non-starter players regarding the playing time during the competition and to evaluate the physical demands of the compensatory training for substitute elite players.

## 2. Materials and Methods

### 2.1. Sample

Substitute football players’ performances from the professional Spanish LaLiga club were analyzed. Three seasons’ games were recorded (2016/2017 (second division); 2017/2018 and 2018/2019 (first division)). A total number of 1047 observations were included in this study. Goalkeepers, players who participated for less than 5 min, substitutes who played since the first half and starter players who were substituted were excluded. Players were divided into starters (*n* = 763) and 3 groups of non-starters (*n* = 285), depending on the minutes played: 5–15 min (*n* = 68); 15–30 min (*n* = 141) and 30–45 (*n* = 75). All players were informed about the objectives and risks and signed a consent form to participate in this investigation. The study protocol was approved by the Local Ethics Committee (Toledo Hospital, Toledo, Spain) and in accordance with the Code of Ethics of the World Medical Association (Declaration of Helsinki).

### 2.2. Procedures

Seasons 2016/2017 (second division), 2017/2018 and 2018/2019 (first division) games and the compensatory training the day after the match (MD + 1C) of a professional football team were analyzed with Apex GPS 10 Hz global positioning system (GPS) (STATSports, Newry, N. Ireland), which had been previously validated [20]. This device was situated at the upper back in a vest that was well-adjusted to the body. It provides data on the time of satellite tracking devices and location, and it receives information that determines the signal traffic. The quality of the signal could change depending on the location and environmental obstruction, and data are more accurate together with the addition of triaxial accelerometers, magnetometers and gyroscopes. At least four satellites are required to determine the GPS position trigonometrically. Apex GPS 10 Hz showed distance bias of 1.05 ± 0.87%, 2.3 ± 1.1% and 1.11 ± 0.99% in the 400 m trial, 128.5 m circuit and 20 m trial, respectively, and a V_peak_ bias of 26.5 ± 2.3 km h^−1^. Data obtained from GPS were downloaded and further analyzed by the STATSports Apex Software (Apex, Brampton, ON, Canada, 10 Hz version 2.0.2.4) [20].

The following variables were collected for all players: the total distance (TD; m) and the distance covered by the player in different high-speed zones: distance in zone 4 (distance covered by the player between 14 and 21 km·h^−1^; m); distance in zone 5 (distance covered by the player between 21 and 24 km·h^−1^; m); distance in zone 6 (distance covered by the player above 24 km·h^−1^; m); peak of maximum velocity (V_MAX_; km·h^−1^); average speed (V_MEAN_; km·h^−1^); and number of actions above 24 km·h^−1^ (number of sprints; *n*). All variables were calculated in absolute and relative terms per minute of play.

The compensatory training session the day after the game for substitute players was included in this study (MD + 1C). The objective of this session was to replicate the competition load for those players who had not completed the match [21]. The starting players performed an active recovery session (MD + 1R), but this session was not analyzed because it was not associated with substitute players.

### 2.3. Statistical Analysis

Data are presented as means ± standard deviations. Kolmogorov–Smirnov distribution test was performed to confirm a normal distribution of the variables and Levene’s test to evaluate the homogeneity of the variance. One-way analyses of variance (ANOVA) were performed to analyze the differences between the titular players and the three groups of substitutes. The post hoc analysis was adjusted by the Games–Howell test because the variances were unequal. Secondly, independent-samples *t*-tests were used to compare the results between divisions for each group of substitutes. Finally, independent-samples t-tests were also used to compare the demands between the titular players of the match and the post training of substitutes. All data were statistically analyzed using SPSS V24.0 for Windows (SPSS Inc., Chicago, IL, USA). The level of significance was set at *p* < 0.05.

## 3. Results

The results revealed, in relative terms (per minute), differences in distances covered and HIAs between the substitute players and the starters (Table 1; *p* < 0.05). The starting players showed lower total distances covered compared to the substitutes who played 5–15 min (−16.77 m·min^−1^; CI95%: −30.09 to −3.44; ES: 0.62), 15–30 min (−6.38 m·min^−1^; CI95%: −9.90 to −2.86; ES: 0.46) and 30–45 min (−3.87 m·min^−1^; CI95%: −7.12 to −0.62; ES: 0.34). However, maximum velocity reached during the game was higher for those who played the whole game (vs. 5–15 min (+1.91 km·h^−1^; CI95%: 1.11 to 2.71; ES: 0.93); vs. 15–30 min (+1.04 km·h^−1^; CI95%: 0.58–1.50; ES: 0.57)). Regarding the minutes played by the substitutes, the analysis of variance showed greater distances covered in zone 4 (14–21 km·h^−1^) for the players with fewer minutes (5–15 min) compared to the distance accumulated by the players who played 15–30 min (+5.18 m·min^−1^; CI95%: 0.53–9.82; ES: 0.48), 30–45 min (+7.28 m·min^−1^; CI95%: 2.62–11.93; ES: 0.75) or starters (+10.32 m·min^−1^; CI95%: 5.96–14.69; ES: 1.08). Starting players covered fewer distances above 24 km·h^−1^ (vs. 5–15 min (−2.49 m·min^−1^; CI95%: −3.96 to −1.03; ES: 0.82); vs. 15–30 min (−1.67 m·min^−1^; CI95%: −2.26 to −1.07; ES: 0.80); vs. 30–45 min (−1.12 m·min^−1^; CI95%: −1.79 to −0.46; ES: 0.61)) and a lower number of sprints (vs. 5–15 min (−0.11 n·min^−1^; CI95%: −0.17 to −0.04; ES: 0.79); vs. 15–30 min (-0.09 n·min^−1^; CI95%: −0.11 to −0.06; ES: 0.86); vs. 30–45 min (−0.06 n·min^−1^; CI95%: −0.09 to −0.03; ES: 0.65)) compared with substitute players (*p* < 0.05).

Regarding the category, non-starters who played 30–45 min in the second division reported higher peak of maximum velocity than in the first division (+1.18 km·h^−1^; CI95%: 0.23–2.13; ES: 0.61; *p* < 0.05). No difference was observed between league standards in any of the other variables (*p* > 0.05; Figure 1).

The compensatory training session (MD + 1C) for substitutes reported significant lower values in all variables compared with starters who played the complete game, especially in sprint distance (−230.48 m; CI95%: −244.08 to −216.88; ES: 2.29; *p* < 0.05). Absolute values revealed a significant decrement of physical demands in non-starters with fewer minutes of play during the game (*p* < 0.05; Table 2).

## 4. Discussion

In the present study, 10 Hz GPS devices (Apex GPS, STATSports, Newry, N. Ireland), have been used to collect all performance data for substitutes during games and MD + 1C, because these units are capable of measuring the smallest changes in acceleration and deceleration, whereas, lower frequency rate units are unable [22,23]. The substitutes performed, in relative terms, higher physical actions compared with starters, and those who played the shortest period of time during the game (5–15 min) gave the highest physical performance compared with the other substitutes who played for longer times (15–30 and 30–45 min). Thus, the less time the non-starters play in the game, the higher relative physical performance they reach. This information will be useful to plan more individualized training sessions adapted to each substitute according to the time of participation during the match, and avoid the same stimulus for all substitutes, therefore, incorrect training load for each player.

Fewer data exist regarding the responses of substitutes entering the field of play [10]. It is difficult to compare our data among substitutes’ time-play with the literature because, to our knowledge, no studies have investigated non-starters’ physical demands regarding the time-play during the competition. Although the time-play of substitutes has not been considered, Bradley et al. [14] mentioned that the players introduced must be immediately able to perform at equivalent or higher work rates than either the players being replaced, others remaining on the pitch or both. In accordance with our investigation, several studies noticed that elite substitutes who had been introduced during the second half covered 25% more HIR and 63% greater sprinting distances during the final 15 min of a game, compared to whole-match players over the same period [23,24,25]. In contrast to our findings, which demonstrated that the shortest time on the pitch is related to a higher physical performance in relative terms, preliminary investigations have indicated an inability of players introduced as substitutes to exceed the running performance that they typically adopt during the first half of matches that they start [24]. Thus, non-starters perform greater physical actions when they are introduced in the game compared with starters.

It is important to notice that GPS devices used in this study have been demonstrated as a valid method to measure distances at low and moderate but not high intensity [19].

Another aim of this study was to analyze the differences between league standard in HIAs of non-starters. No differences were found between first and second divisions regarding physical performance of non-starters. These results are in contrast with several researches that have shown that players at a higher standard of play perform more high-intensity running than peers at lower standards [24,26,27]. In the elite Italian League, players performed 28% more high-intensity running than sub-elite Danish League peers [24]. However, the data were captured from two separate European Leagues of vastly different standards. Similarly, Ingebrigtsen et al. [27] reported that distance covered in high-intensity running was 30–40% greater in players in top versus middle- and bottom-ranking Danish teams. However, in contrast with these findings, Di Salvo et al. [28] showed that Championship players covered greater distances in jogging, running, high-speed running and sprinting than Premier League players. Another similar investigation found that players in League 1 and the Championship performed more high-intensity running than those in the Premier League. Players also covered more high-intensity running when moving down from the Premier League to the Championship but not when players moved up standards [29]. All these studies focused on starters’ performance and are compared with our findings that refer to non-starter players because, to our knowledge, no investigation has analyzed the differences in physical performance of substitute players across two standards of elite football within a single country.

The present study also evaluated the physical demands of the compensatory training (MD + 1C) for substitutes. Players with reduced game time will require a training session that replicates competition loads, whereas those players completing the game will require a recovery session instead (MD + 1R) [17]. Our results found that MD + 1C load is lower than competition load. Accordingly, Stevens et al. [17] noted that non-starters training on the day after the match showed significantly lower values than starters on total distance (−17 ± 5%), energy expenditure (−18 ± 5%), time spent above 90%HRmax (−52 ± 35%), running (−55 ± 13%), accelerations (−42 ± 12%), decelerations (−46 ± 15%) and high power (−30 ± 9%), contributing on average to a lower estimated total load. In contrast, Martín García et al. [21] showed that MD + 1 exceed 50% of match play values, and these included total distance covered (53%), average metabolic power (69%), accelerations (86%) and decelerations (80%), and could be an ideal day to compensate for the reduced competition load in players with limited game time. Moreover, the present study found that the accumulation of match game load and MD + 1C training session for non-starters are lower than full game load for starters. In line with this, the match was the most demanding session of the week only for the starters, contributing to considerably lower values for substitutes’ training [17], physical fitness performance decrements [19] and a general risk of under-loading non-starters having injuries [30]. Thus, competition time is the main source of differences between starters and non-starters in accumulated training load [9]. These results are important for coaches in order to stimulate substitutes players with additional training load until reaching the full-game accumulation load. Moreover, the game time played by each substitute monitored will help to individualized training load in different groups according this time of participation during the competition.

In absolute terms, non-starters’ load accumulation depends on the game time during the competition and it is difficult to compare with other studies because no studies have separated time-play periods of substitutes. Altogether, there is clearly a challenge to sufficiently load non-starters within the specific and individual context of each player. Further research is needed to refine training prescription of compensatory training sessions for non-starters to ensure their readiness for competition, considering specific game time played.

## 5. Conclusions

The present study demonstrates that substitutes players performed higher physical actions at high-intensity (total distance, accelerations, number of sprints, V_max_, V_mean_) per minute compared with starters. Moreover, substitutes who played the shortest period of time during the game (5–15 min) produced the highest physical performance compared with the other substitutes who played more time (15–30 and 30–45 min). Therefore, substitutes’ game loads depend on the time of playing. Regarding the league standard, the physical performance of substitutes was not significantly different between first and second divisions. Additionally, compensatory training (MD + 1C) accumulated training load has been demonstrated as lower comparing with the full game, showing that substitutes are at risk of being undertrained, have a reduced physical performance and increased risk of injury. These results increase the knowledge for coaches to plan training sessions according to the time-play of each substitute and guarantee a more individualized training load which will aid to decrease injuries and optimize performance during the competition.

## Figures and Tables

**Figure 1 sensors-22-03872-f001:**
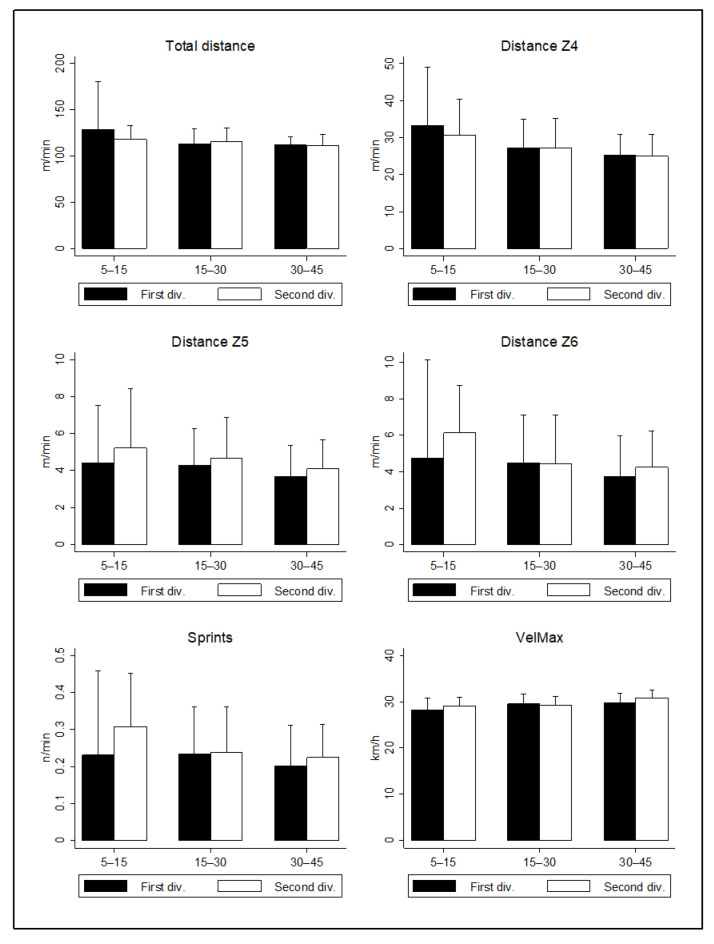
Distances covered and high-intensity actions by substitutes who play 5–15 min, 15–30 min, 30–45 min. Differences between the 1st and 2nd divisions.

**Table 1 sensors-22-03872-t001:** Distances covered and high-intensity actions by substitutes who play 5–15 min (a), 15–30 min (b), 30–45 min (c) and starters (d).

	5–15 Min (a)	15–30 Min (b)	30–45 Min (c)	Starter (d)
Total Distance (m·min^−1^)	124.78 ± 41.57 ^d^	114.39 ± 15.22 ^d^	111.88 ± 10.04 ^d^	108.01 ± 12.41 ^a,b,c^
Distance Zone 4 (m·min^−1^)	32.36 ± 13.57 ^b,c,d^	27.19 ± 7.83 ^a,d^	25.09 ± 5.79 ^a,d^	22.04 ± 5.49 ^a,b,c^
Distance Zone 5 (m·min^−1^)	4.73 ± 3.13 ^d^	4.44 ± 2.09 ^d^	3.83 ± 1.65 ^d^	2.86 ± 1.11 ^a,b,c^
Distance Zone 6 (m·min^−1^)	5.29 ± 4.56 ^d^	4.46 ± 2.65 ^d^	3.92 ± 2.14 ^d^	2.80 ± 1.52 ^a,b,c^
Number of sprints (n·min^−1^)	0.26 ± 0.20 ^d^	0.24 ± 0.12 ^d^	0.21 ± 0.10 ^d^	0.15 ± 0.07 ^a,b,c^
V_MEAN_ (km·h^−1^)	9.59 ± 0.91 ^d^	9.51 ± 0.67 ^d^	9.29 ± 0.57 ^d^	8.79 ± 0.55 ^a,b,c^
V_MAX_ (km·h^−1^)	28.55 ± 2.46 ^c,d^	29.41 ± 1.99 ^d^	30.10 ± 2.04 ^a^	30.45 ± 1.65 ^a,b^

Data are mean ± SD; ^a,b,c,d^ Significant differences between groups (*p* < 0.05).

**Table 2 sensors-22-03872-t002:** Absolute distances covered and high-intensity actions during compensatory training for substitutes (MD + 1C), for starters on match day and non-starters depending on the minutes of play.

	MD + 1C	MD Starters	MD 5–15 min NS	MD 15–30 min NS	MD 30–45 min NS
Total Distance (m)	4654.89 ± 919.59 *	10,171.23 ± 955.22	1310.50 ± 463.35 *	2495.90 ± 571.02 *	3841.76 ± 468.72 *
Distance Zone 4 (m)	447.44 ± 330.92 *	2075.72 ± 508.68	339.10 ± 153.63 *	589.95 ± 188.05 *	857.94 ± 190.93 *
Distance Zone 5 (m)	71.33 ± 63.4 *	269.60 ± 103.05	49.00 ± 31.50 *	95.56 ± 45.47 *	130.55 ± 54.50 *
Distance Zone 6 (m)	33.41 ± 57.64 *	263.90 ± 143.70	53.47 ± 43.24 *	96.19 ± 56.97 *	134.39 ± 72.84 *
Number of sprints (*n*)	3.96 ± 3.74 *	14.25 ± 7.08	2.62 ± 1.90 *	5.07 ± 2.61 *	7.17 ± 3.47 *
V_MEAN_ (km·h^−1^)	3.84 ± 0.59 *	8.79 ± 0.55	9.59 ± 0.91 *	9.51 ± 0.67 *	9.29 ± 0.57 *
V_MAX_ (km·h^−1^)	26.48 ± 3.23 *	30.45 ± 1.65	28.55 ± 2.46 *	29.41 ± 1.99 *	30.10 ± 2.04 *

Data are mean ± SD; MD: match day; NS: non-starters; V_MEAN_: mean speed; V_MAX_: maximum speed; * significant differences with MD starters (*p* < 0.05).

## Data Availability

Not applicable.

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
