# Peer review of "Physical Demands during the Game and Compensatory Training Session (MD + 1) in Elite Football Players Using Global Positioning System Device"

_sensors, 2022, doi:10.3390/s22103872_

Round 1
Reviewer 1 Report
The study titled Physical Demands During the Game and Compensatory Train-2 ing Session (MD+1) in Elite Football Players Using Global Posi-3 tioning System Device is a very interesting paper about the physical demands of the compensatory training (MD+1C) for substitute players in elite football. The text is clear and is well written. Nevertheless, there are some suggestion I would like to declare.
Abstract
In my opinion the first part of the first sentence in not adequate for the study “The aims of this study were to use tracking system device to quantify”, because the focus of the paper is not to use of the technology. This is just a technique the authors used to achieve the aim. Please revise.
Maybe, the argument of the papers must be oriented to compare if the MD+1 is really compensatory. Therefore, it is not a question of comparing the players but of comparing the training loads considering the sum of the 'partial' competition demands and the DM+1
Introduction
The objective “Therefore, the purpose of this study was to analyse the differences of high-intensity physical demands of Non-Starters…” must be re-written because not only the high-intensity variables are going to be analised.
Sample
Here two question about two decisions done. Why substitutes who played since the first half were excluded? Why has the authors make the decision to grouped the substitutes in periods of 15’? the proposal I want to explain is regarding to maybe it would be very interesting to accumulate the demands from both the match and the MD+1.
Please, include information about the average and sd of the participation of the starters and no-starters and the same for each group pf players.
Procedures
I do not understand why the authors use absolute values. The participation of the players are not the same then the absolute values are not comparable. In this line is very logical the information appears in the table 2, “more time training or playing more demands”.
Results
Line 156. I think there is a mistake, because the zone 4 is not between 18-21 Km/h.
Figure 1 is not necessary
Tables 1 and 2. The values of the starters need more explanation because the participation of this type of players would be very different.
Probably there was a large range of min played by players in matches, so use the ranges of 15 min maybe is not the best option. The players were grouped in the same group players with 15 min and with 30. Then the values of the demands from the absolute point of view could have some problems.
The average value of the MDstarters is around 10 Km, this value is just of the players that finished the match or are all the starters included, also the players who played 46’?
Discussion
Please reduce the information of the first paragraph. It is vey large and are ideas that are repeated.
Lines 187-192 must be remove, it is not the aim of the study. The use of technology is just a process.
This affirmation (line 207) can not be said “In contrast to our findings, which demonstrated that the shortest time on the pitch guarantees a higher physical performance”, please rewrite “guarantees”, because it is not mandatory, situational variables have a very high impact in these demands.
This sentence or idea is repeated, remove please: “A novel aspect of this study was that substitutes who played the shortest period of time during the game (5–15 minutes) produced the highest physical performance compared with the other substitutes who played more time (15–30 and 30–45 minutes).”
Again, in my opinion, the paragraph (from line 225 to 231) is not necessary to highlight in this study
Reviewer 2 Report
The manuscript entitled “Physical Demands During the Game and Compensatory Training Session (MD+1) in Elite Football Players Using Global Positioning System Device” by Gabriel Calderón-Pellegrino et.al present a study using tracking system device to quantify the demands of substitutes with regard to the playing time during the competition and to evaluate the physical demands of the compensatory training (MD+1C) for substitute football players. Although there is some confusing expression, the study is of significance for future study.
- Line 41, please check whether “(Witte & Wilson, 2004)” should be deleted.
- Line 70, please explain the meaning of “match day -4”.
- It would be better if Figure 1 was placed in 2.2 rather than 3.
